

# The functional analysis of ABCG transporters in the adaptation of pigeon pea (*Cajanus cajan*) to abiotic stresses

Lili Niu[1,2,*], Hanghang Li[1,2,*], Zhihua Song[1,2], Biying Dong[1,2], Hongyan Cao[1,2], Tengyue Liu[1,2], Tingting Du[1,2], Wanlong Yang[1,2], Rohul Amin[1], Litao Wang[1], Qing Yang[1,2], Dong Meng[1,2] and Yujie Fu[1,2,3]

[1] The College of Forestry, Beijing Forestry University, Beijing, People's Republic of China
[2] Beijing Advanced Innovation Center for Tree Breeding by Molecular Design, Beijing, People's Republic of China
[3] Key Laboratory of Forestry Plant Ecology, Ministry of Education, Northeast Forestry University, Harbin, People's Republic of China
[*] These authors contributed equally to this work.

Corresponding authors
Dong Meng, mengdongjlf@163.com
Yujie Fu, yujie_fu@163.com

## ABSTRACT

ATP-binding cassette (ABC) transporters are a class of proteins found in living organisms that mediate transmembrane transport by hydrolyzing ATP. They play a vital role in the physiological processes of growth and development in plants. The most numerous sub-type transporter in the ABC transporter family is the ABCG group and which have the most complex function in a plant's response to abiotic stresses. Our study focused on the effect of ABCG transporters in the adaptation of the pigeon pea to adverse environments (such as drought, salt, temperature, etc.). We conducted a functional analysis of ABCG transporters in the pigeon pea and their role in response to abiotic stresses. A total of 51 *ABCG* genes (*CcABCGs*) were identified, and phylogenetic analysis was conducted. We also identified the physicochemical properties of the encoded proteins, predicted their subcellular localization, and identified of the conserved domains. Expression analysis showed that *ABCG* genes have different expression profiles with tissues and abiotic stresses. Our results showed that *CcABCG28* was up-regulated at low temperatures, and *CcABCG7* was up-regulated with drought and aluminum stress. The initial results revealed that ABCG transporters are more effective in the abiotic stress resistance of pigeon peas, which improves our understanding of their application in abiotic stress resistance.

## INTRODUCTION

ATP-binding cassette (ABC) transporters are a part of the largest and oldest known protein families and are widely found in eukaryotic and prokaryotic organisms (*Martinoia et al., 2002*; *Mosser et al., 1993*). ABC transporters play a crucial role in the growth and development of plants by detoxifying exogenous toxins in response to abiotic stress, and transporting metabolites including intercellular peptides, sugars, lipids, alkaloids, inorganic ions and other metabolic substances (*Mendez & Salas, 2001*; *Morris & Zhang,*

*2006*; *Mourez, Hofnung & Dassa, 1997*). *AtABCB1* (also known as *AtMDR1*) was first identified in *Arabidopsis* in 1992 (*Dudler & Hertig, 1992*). ABC transporters are divided into eight subfamilies in the plant genome (ABCA-ABCG and ABCI) according to the Human Genome Organization (HUGO). There are 129 ABC transporters identified in Arabidopsis genomes (*Arabidopsis thaliana*), 128 identified in rice genomes (*Oryza sativa*), and 261 identified in soybean (*Glycine max*) genomes (*Mishra et al., 2019*; *Schulz & Kolukisaoglu, 2006*; *Sanchez-Fernandez et al., 2001*).

The ATP-binding cassette subfamily G is the largest subfamily of the ABC transporter family. The ABCG protein is widely distributed in plants and plays an important role in many fundamental physiological processes (*Kretzschmar et al., 2011*). Highly conservative amino acids, including 1–2 nucleotide-binding domains (NBDs), a highly hydrophobic transmembrane domain, and 1-2 trans-membrane domains (TMDs) are typical structural features of ABCG transporters (*Verrier et al., 2008*). The ABCG subfamily was divided into white-brown complexes (WBCs) and pleiotropic drug resistance (PDRs) complexes. WBCs are semi-molecular ABCG transporters that contain one NBD domain and one TMD domain. PDRs belong to the full-molecule ABCG transporter and include two NBD domains and two TMD domains (*Jasinski et al., 2009*). The NBD domain consists of three highly conserved regions of approximately 200 amino acids and include the Walker A box [GX$_4$GK (ST)], Walker B box [(RK) X$_3$GX$_3$L (hydrophobic)$_3$], and Walker C and are approximately 120 amino acids in length (*Walker et al., 1982*). The TMD domain contains 4-6 $\alpha$-helixes, which are the channel for the transport of substrate molecules for trans-membrane transport (*Hyde et al., 1990*; *Schneider & Hunke, 1998*). It is generally believed that the mechanism of ABCG transporters occurs on the substrate recognition site on the TMD domain and works to recognize and bind transport substrates located near the cell membrane. The NBD domain on the cell membrane hydrolyzes ATP to provide energy for substrate transport and causes conformational changes in the membrane structure (*Davidson & Maloney, 2007*). *SpTUR2*, the first full-molecule *ABCG* transporter gene in plants, was identified in the perennial aquatic plant *Spirodella polyrhiza* in 2002 (*Van den Brule et al., 2002*). More than 40 ABCG transporters have been identified in *Arabidopsis*, rice, and soybean genomes, to date. ABCG transporters are involved with the plant's many physiological activities. For example, *OsABCG31* in rice may decrease evaporation from plant leaves and may be related to the drought stress response of rice (*Chen et al., 2011*). *Arabidopsis AtPDR8* is involved with Na$^+$ excretion, increasing the plant's tolerance to salt and drought stress (*Kim et al., 2007*). *AtPDR36* also responds to the toxic heavy metal effects in *Arabidopsis thaliana* and participates in the stomatal self-regulation in leaves (*Kim et al., 2010*).

*Cajanus cajan* (L.) Millsp, also known as pigeon pea, is a diploid plant ($2n = 22$) with a genome size of approximately 858 Mbp. It grows in tropical and subtropical regions and has a stable regulatory system to adapt to conditions that include high temperature, high salinity, and drought (*Singh et al., 2013*; *Varshney et al., 2011*; *Wu et al., 2011*; *Yadu et al., 2018*). Cooler environmental temperatures have led to the northern migration of the pigeon pea. The pigeon pea grows easily in acidic soil (pH 5-7) and is resistant to aluminum, which has bought it much attention. The numerous secondary metabolites

of the pigeon pea play an important role in the adaptation of the plant to its adverse environment (*Shepherd & Bhardwaj, 1986*). It also possesses medicinal properties and is used widely in the chemical industry (*Ogoda, Akubue & Okide, 2002*; *Pandey & Pandey, 1991*).

Our research focuses on the trans-membrane transport of secondary metabolites, antibiotics, and heavy metal ions by ABCG transport protein hydrolysis of ATP (*Yazaki, 2006*; *Badri et al., 2008a*; *Badri et al., 2008b*; *Le Hir et al., 2013*; *Fourcroy et al., 2014*). The inhibition of ABCG transporters decreases plant flavonoid content (*Morris & Zhang, 2006*; *Imai et al., 2004*). Plants that transport some hormones (such as ABA) can improve their survivability in drought and other adverse conditions (*Kuromori et al., 2010*; *Kang et al., 2010*). ABCG transporters could positively affect a plant's response to adversity by transporting specific substances. We identified family genes, and conducted phylogenetic and expression analyses to reveal the important role of ABCG transporters in resisting environmental stress in the pigeon pea. A total of 51 ABCG transporters were identified and further analyzed. We found cis-acting elements related to the stress response in several genes identified above, suggesting that the ABCG transporter had a related regulatory effect in the pigeon pea. We analyzed the expression of the ABCG gene in different organs and different stress treatments and observed the tissue specificity of the ABCG gene and stress expression response to explore the expression profile of the ABCG gene in the pigeon pea. Our results provide scientific support for exploring the mechanism of ABCG transporters in relation to the resistance of several abiotic stressors of the pigeon pea.

## MATERIALS AND METHODS

### Identification of ABCG transporters in the pigeon pea genome

One hundred-twenty nine *Arabidopsis* ABC protein sequences were downloaded from the Phytozome v12.1 database (https://phytozome.jgi.doe.gov/pz/portal.html). All of the *Arabidopsis* ABCs were used to identify the ABC transporters in the *Cajanus cajan* (L.) Millsp (taxid: 3821) database using BLASTP search in the National Center for Biotechnology Information (NCBI) with an initial cut-off e-value of $1.0\ e^{-10}$ and max target sequence of 500. The Hidden Markov Model (HMM) profiles of ABC transporters (such as PF00005, PF00664, and PF10614) were downloaded from the Pfam database (http://pfam.xfam.org/search#tabview=tab1) and the HMMER search server was used against the pigeon pea proteome with an $E$-value setting of $1.0\ e^{-5}$ (https://www.ebi. ac.uk/Tools/hmmer/) (*Potter et al., 2018*). The resulting protein sequences were further identified by a conserved domain of ABC based on a conserved domain search (CD-search) on the NCBI website with a threshold e-value of $1.0\ e^{-5}$. We identified members of the ABC transporters using the above approach.

Multiple-sequence alignments were performed on the ABC transporters identified above and several ABC proteins of *Arabidopsis thaliana*, rice, and soybean, using clustalW with default settings. The ABC transporters' phylogenetic tree was constructed using the neighbor-joining (NJ) method with 1000 replications of bootstrap and p-distance of a

model in MEGA6.0 (https://www.megasoftware.net/) (*Tamura et al., 2013*). The phylogenetic tree was visualized using the iTOL website (http://itol.embl.de/help.cgi) (*Sugiyama et al., 2011*). Finally, ABCG transporters were identified through the phylogenetic analysis of the ABC family of the pigeon pea.

## Phylogenetic tree construction and chromosome localization

The *ABCG* genes' chromosomal information was obtained through the NCBI website (https://www.ncbi.nlm.nih.gov/) and were systematically named based on the number and relative position of the *ABCG* genes in the pigeon pea chromosome. The phylogenetic tree of the pigeon pea ABCG transporters was constructed in MEGA6.0, as described above.

The chromosomal location of the *CcABCGs* was mapped using the MapChart software (https://www.wur.nl/en/show/Mapchart.htm) based on the chromosomal location information of the ABCG transporters (*Voorrips, 2002*).

## Protein property prediction, subcellular localization

The exon information of *CcABCGs* was collected on the NCBI website. The number of amino acid sequences, relative molecular weight, and the isoelectric point (pI) of CcABCGs was predicted in the ExPASy ProtParam server database (http://expasy.org/). We predicted the subcellular localization of ABCG proteins using the Cell-PLoc 2.0 database (http://www.csbio.sjtu.edu.cn/bioinf/Cell-PLoc-2/) (*Chou & Shen, 2008*).

To verify our predictions of subcellular localization, two-week-old pigeon pea seedlings were used as a cDNA template to amplify the full-length coding region of the *CcABCG7* gene. The primers were: F: 5′-ATGGTGATGATATGGGAAAATGTTAC- 3′ and R: 5′-TTATATTGGAAGGTTTGGGGACA- 3′. The recovered PCR production was ligated to the T vector pMD19-T (TaKaRa, Japan). Subsequently, it was cloned into the eGFP-pROK II vector by double digestion using Kpn I/Xb I to construct the CcABCG7-eGFP-pROK II expression vector, and was transformed into the agrobacterium strain GV3101 (Shanghai Weidi Biotechnology, China). One-month-old tobacco seedlings were injected into the agrobacterium liquid on the back of tobacco leaves with a disposable syringe and placed in the dark for 3 days for observation. Approximately 0.5 cm$^2$ of the material was taken around the injection site, observed, and photographed under a Leica SP8 laser confocal microscope (Leica, Germany).

## Analysis of motifs and conserved domain

To examine the characteristics and properties of the pigeon pea ABCG subfamily protein domain, the conserved motif of CcABCGs was analyzed on the MEME software http://meme-suite.org/tools/meme with following parameters: the number of motifs was set 10 and the optimum motif width was set between 20 and 200, the other parameters were set to their default (*Bailey & Elkan, 1994*). Then InterPro program http://www.ebi.ac.uk/interpro/scan.html was used to annotate all 10 motifs.

The conserved domain of CcABCGs was performed using the HMMER servers (https://www.ebi.ac.uk/Tools/hmmer/) and visualized with TBtools software (*Chen, 2018*). We selected representative WBC and PDR transporter amino acid sequences as target

sequences and chose the template with higher homology and better coverage using the Swiss-model server set to automatic (https://swissmodel.expasy.org/interactive). The model results were evaluated using the SAVES server (https://servicesn.mbi.ucla.edu/SAVES/), and the homology model of the CcABCG proteins was visualized by the PyMol software.

## Gene structure and cis-elements analysis of CcABCGs

The intron/exon structures of the *ABCG* genes were analyzed using the gene annotation file (GFF), C.cajan_V1.0, and visualized using TBtools software (*Chen, 2018*), based on the evolution analysis of *CcABCGs*. The 2,000 bp upstream region of all identified *CcABCGs* was extracted from the pigeon pea genome to identify the stress-related or other functional cis-acting regulatory elements of the promoter sequences using TBtools software. All promoter sequences were analyzed using PlantCARE software (http://bioinformatics.psb.ugent.be/webtools/plantcare/html/) (*Rombauts et al., 1999*).

## GO annotation and function prediction

GO (Gene Ontology) integrates and unifies the description and standards of gene products in a database and provides the most comprehensive description of gene functions and gene products. The ABCG protein sequences in the pigeon pea were used for blast alignment in the Swiss-prot database (https://www.uniprot.org/blast/) and the alignment results were annotated and classified by the pigeon pea *ABCG* gene (*Ashburner et al., 2000*; *Consortium, 2019*; *Ehlert et al., 2006*; *Yang et al., 2016*).

## Plant materials and treatments

Pigeon pea seeds (ICPL87119) were sown in a soil mixture of nutrient soil, vermiculite, and perlite (1:1:1) in the greenhouse at Beijing Forestry University in China with natural light for 10-14 h/day, a temperature of $18-28\,°C$ and a relative humidity of 45–70%. After three months, the roots, stems, leaves, and flowers from pigeon pea trees were selected and stored at $-80\,°C$ to extract RNA and further analyze their tissue-specific expression.

The pigeon pea seeds were surface-sterilized with 75% ethanol for 30 s, then soaked in sodium hypochlorite solution for 6 min, and finally washed 5 times with sterile water for 30 s durations. These sterilized seeds were grown in Murashige and Skoog (MS) medium at pH=5.8 in growth chambers at $24 \pm 2\,°C$ with a photoperiod of 16 h light/8 h dark, $400\,\mu M\,m^{-2}\,s^{-1}$ light intensity. To analyze the expression of the *CcABCG* genes in various abiotic stresses, the 14-day old pigeon pea seedlings were assigned to different treatments. The treatments were as follows: heat and cold stress, in which the seedlings were cultured at $4\,°C$ and $42\,°C$ in incubators, respectively; salt and metal stress, in which the seedlings were grown in solid MS medium with 200 nmol/L NaCl and $100\,\mu mo/L$ $AlCl_3$, respectively; drought stress, in which the pigeon pea seedlings were treated with 250 nmol/L mannitol in solid MS medium. The roots and leaves of these seedlings were sampled at 0 h, 6 h, and 12 h after various treatments. Three biological replicates of each sample were immediately frozen in liquid nitrogen and stored at $-80\,°C$ until expression analysis of variously abiotic stresses was conducted.

### RNA isolation and quantitative real-time PCR analysis

We selected 10 genes containing elements that responded to adversity for tissue specificity analysis and abiotic stress analysis based on the above analysis of cis-elements in the promoter region. Total RNA was isolated using the CTAB method and first-strand cDNA was synthesized from 1 µg of total RNA using a PrimeScript RT kit (Takara) according to the manufacturer's instructions. The quality and concentration of cDNA were assessed using a Nano Photometer N50 (Implen GmbH, Munich, Germany). We performed expression analysis of the ABCG transporter under different tissues and abiotic stresses using qRT-PCR. Real-time RT-PCR analysis was performed using CFX connect (Bio-Rad, California, USA) with the SYBR Green PCR Master Mix (TaKaRa, Tokyo, Japan) with *CcActin* as a reference gene. The gene primers selected were synthesized on the Sangon Biotech website (http://www.sangon.com/newPrimerDesign) and are shown in Table S1. All analyses were performed with three biological replicates and qRT-PCR data was analyzed using IBM SPSS 22 software (IBM Corporation, USA).

## RESULTS

### Identification of ABCG transporter genes in the pigeon pea genome

One hundred-twenty nine Arabidopsis ABC proteins were downloaded and used as queries in the pigeon pea genome database using BLASTP server to identify all members of the pigeon pea ABCG transporters. A total of 222 ABC proteins were identified in the pigeon pea after removing redundancies. The HMMER search was performed using the HMM of ABC transporters (such as PF00005, PF00664, and PF10614) to screen for ABC transporters identified by BLASTP. One hundred fifty-five ABC proteins were eventually confirmed to have NBD/TMD domains (Table S2). To explore the phylogenetic evolutionary relationship between the pigeon pea and other species including Arabidopsis, rice, and soybean, a Neighbor-Joining (NJ) tree was constructed (Fig. S1) based on ABC proteins. A phylogenetic relationship indicated that the pigeon pea *ABC* gene family was classified into 8 different groups (ABCA-ABCI), and the ABCG subfamily, which contained 51 members, was the largest group of ABC transporters in the pigeon pea.

### Phylogenetic tree construction and chromosome localization

All of the ABCG transporters identified above were named *CcABCG1-CcABCG51* based on their chromosomal location. A Neighbor-Joining (NJ) tree was constructed to analyze the phylogenetic relationship of ABCG transporters (Fig. 1). Phylogenetic analysis indicated that the ABCG transporters were further divided into WBC and PDR, in which WBC contained 33 ABCG transporters and PDR contained 18 transporters.

The chromosome localization of ABCG transporters was determined in the pigeon pea genome and all identified *CcABCGs* were found to be diversely distributed on pigeon pea chromosomes, except for chromosome 7. Chromosomes 1 and 8 both contained one ABCG transporter and all chromosomes of the pigeon pea had the least ABCG transporters. Chromosomes 2, 9, and 11 included 5, 5, and 8 transporters, respectively. Seventeen genes were distributed on unplaced scaffolds and were mapped into a single chromosome, ChrUn (Fig. 2). Two pairs of genes (*CcABCG8/9, CcABCG32/33*) were
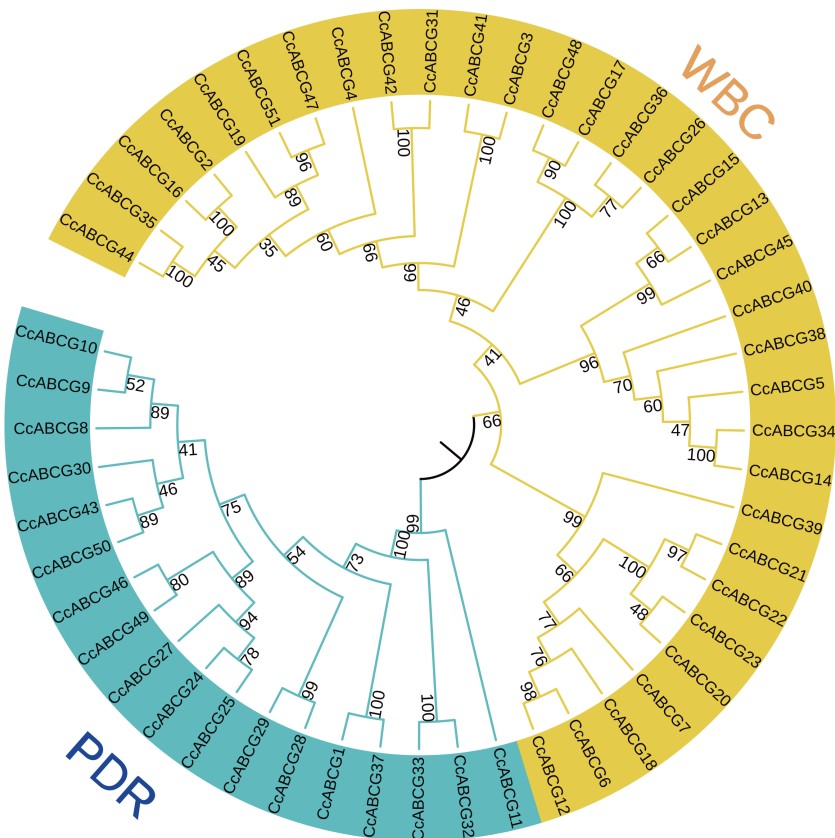

**Figure 1** **Phylogenetic analysis of the ABCG transporters among pigeon pea.** Using the MEGA6.0 program, the NJ (Neighbor-Joining, NJ) tree was constructed using the amino acid sequence of the pigeon pea ABCG transporters. The numbers beside the branches represent bootstrap values based on 1,000 replications. The outer side of the phylogenetic tree is a branch labeled with two subgroups of the pigeon pea ABCG transporters, and shown in different colors. WBC, white-brown complex; PDR, pleiotropic drug resistance.

closely linked on the chromosome and belonged to the paralogous genes. The *ABCG* gene locations were annotated using the NCBI website and the information in shown in Table 1.

## Protein property prediction, subcellular localization of CcABCGs

Fifty-one CcABCGs were identified and their protein properties were predicted (Table 1), including their amino acid lengths, relative molecular weights, and isoelectric points (pIs). Results indicated that the lengths of all ABCG proteins ranged from 609 aa (CcABCG34) to 1500 aa (CcABCG29). Similarly, their relative molecular mass ranged from 68159.95 Kd (CcABCG34) to 170370.83 Kd (CcABCG29) and the theoretical isoelectric points ranged from 6.75 (CcABCG29) to 9.47(CcABCG4). The *ABCG* genes' exon numbers changed from 1 to 26.

We performed the subcellular localization of CcABCGs to determine the active sites of the CcABCGs. The prediction of the subcellular localization indicated that all CcABCGs

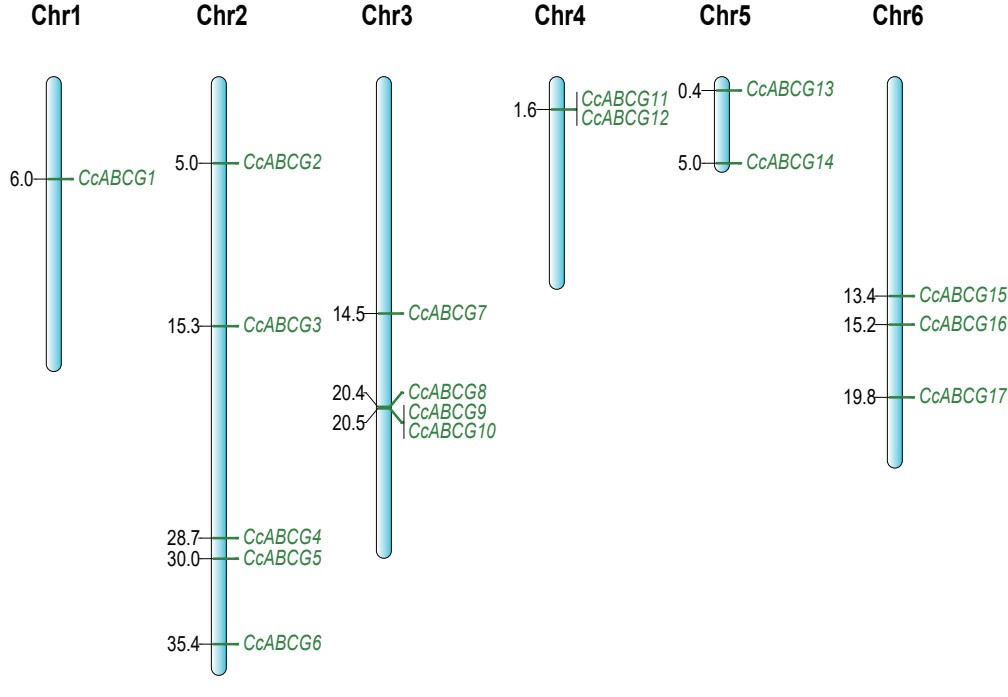

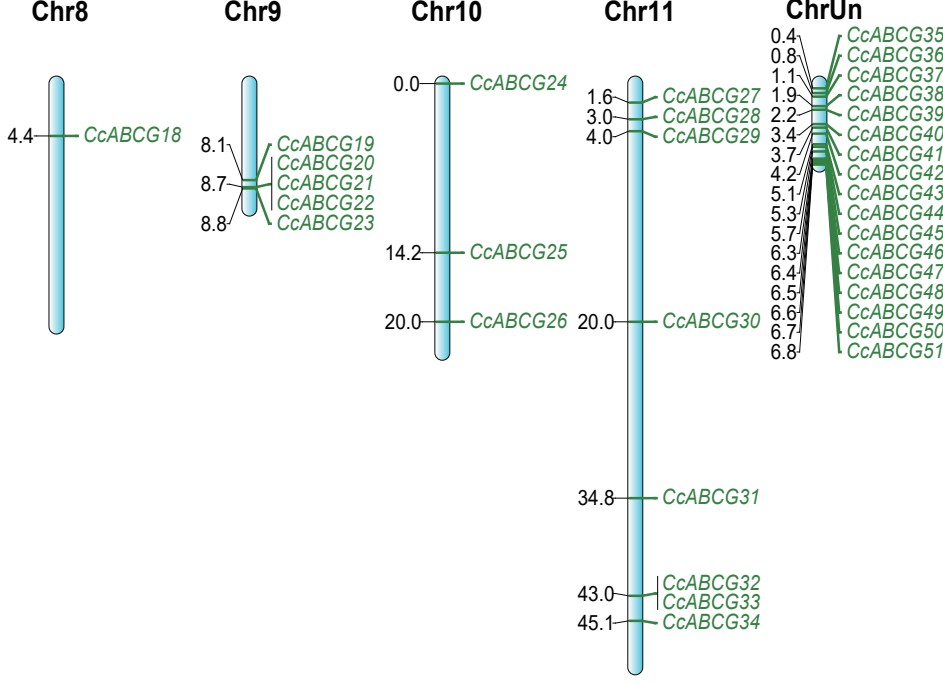

Figure 2 **The chromosomal location of the pigeon pea *ABCG* genes.** The pigeon pea *ABCG* genes were named *CcABCG1-CcABCG51* based on chromosomal location information. All of *ABCG* genes were location in all pigeon pea chromosomal except of chromosomal 7. ChrUn contained 17 members, which were mapped based on the length of those unplaced scaffold sequences with 17 genes, respectively.

**Table 1** Prediction of physicochemical properties and subcellular localization of ABCG transporters.

| Gene | Accession | Location | Position | Length (aa) | Mw | pI | Chr | Exon | Sub-loc |
|---|---|---|---|---|---|---|---|---|---|
| CcABCG1 | KYP76113 | LOC109804364 | 6,006,382–6,022,606 | 1,421 | 159155.33 | 7.76 | 1 | 23 | Cell membrane |
| CcABCG2 | KYP72481 | LOC109789679 | 5,026,814–5,030,449 | 645 | 72010.85 | 9.08 | 2 | 5 | Cell membrane |
| CcABCG3 | KYP73490 | LOC109810734 | 15,320,209–15,323,794 | 684 | 77318.74 | 8.11 | 2 | 10 | Cell membrane |
| CcABCG4 | KYP74721 | LOC109795335 | 28,723,419–28,730,237 | 668 | 73512.6 | 9.47 | 2 | 3 | Cell membrane |
| CcABCG5 | KYP74838 | LOC109806747 | 29,968,224–29,970,344 | 619 | 69012.83 | 9.4 | 2 | 1 | Cell membrane |
| CcABCG6 | KYP75362 | LOC109815813 | 35,437,420–35,445,042 | 726 | 79570.36 | 8.27 | 2 | 12 | Cell membrane |
| CcABCG7 | KYP70503 | LOC109797228 | 14,492,247–14,496,242 | 660 | 73883.51 | 8.81 | 3 | 9 | Cell membrane |
| CcABCG8 | KYP71088 | LOC109796083 | 20,447,829–20,456,456 | 1,448 | 164560.21 | 8.28 | 3 | 20 | Cell membrane |
| CcABCG9 | KYP71089 | LOC109796084 | 20,469,958–20,479,590 | 1,444 | 163478.51 | 8.27 | 3 | 20 | Cell membrane |
| CcABCG10 | KYP71090 | LOC109796805 | 20,488,744–20,497,576 | 1,454 | 164868.63 | 7.95 | 3 | 20 | Cell membrane |
| CcABCG11 | KYP68041 | LOC109798752 | 1,617,897–1,630,220 | 718 | 80600.64 | 9.1 | 4 | 9 | Cell membrane |
| CcABCG12 | KYP68044 | LOC109798893 | 1,641,751–1,647,610 | 686 | 76872.23 | 8.88 | 4 | 10 | Cell membrane |
| CcABCG13 | KYP67437 | LOC109799601 | 355,237–358,794 | 814 | 91841.52 | 8.95 | 5 | 6 | Cell membrane |
| CcABCG14 | KYP67863 | LOC109799748 | 5,000,555–5,003,051 | 643 | 72113.02 | 7.88 | 5 | 1 | Cell membrane |
| CcABCG15 | KYP66379 | LOC109800436 | 13,422,611–13,426,469 | 721 | 81114.25 | 8.97 | 6 | 3 | Cell membrane |
| CcABCG16 | KYP66524 | LOC109801031 | 15,158,990–15,162,297 | 663 | 74678.78 | 8.6 | 6 | 5 | Cell membrane |
| CcABCG17 | KYP66999 | LOC109800523 | 19,843,419–19,853,680 | 1,092 | 121456.99 | 8.87 | 6 | 14 | Cell membrane |
| CcABCG18 | KYP61725 | LOC109804023 | 4,405,565–4,412,603 | 704 | 78466.57 | 8.83 | 8 | 10 | Cell membrane |
| CcABCG19 | KYP61105 | LOC109804765 | 8,122,601–8,126,214 | 632 | 70593.09 | 9.02 | 9 | 4 | Cell membrane |
| CcABCG20 | KYP61155 | LOC109805263 | 8,663,643–8,668,819 | 658 | 73428.07 | 7.99 | 9 | 8 | Cell membrane |
| CcABCG21 | KYP61157 | LOC109805167 | 8,722,014–8,728,825 | 679 | 75362.12 | 8.36 | 9 | 8 | Cell membrane |
| CcABCG22 | KYP61159 | LOC109805166 | 8,746,304–8,752,313 | 681 | 75328.34 | 8.65 | 9 | 8 | Cell membrane |
| CcABCG23 | KYP61160 | LOC109804802 | 8,756,197–8,761,747 | 648 | 71982.39 | 8.81 | 9 | 9 | Cell membrane |
| CcABCG24 | KYP58337 | LOC109805773 | 11,158–17,707 | 1,427 | 161183.4 | 7.36 | 10 | 21 | Cell membrane |
| CcABCG25 | KYP59575 | LOC109805875 | 14,201,296–14,246,400 | 1,431 | 161968.54 | 7.97 | 10 | 21 | Cell membrane |
| CcABCG26 | KYP60108 | LOC109805696 | 19,992,101–20,040,472 | 1,104 | 123273.83 | 8.89 | 10 | 16 | Cell membrane. Chloroplast. |
| CcABCG27 | KYP53996 | LOC109808825 | 1,626,589–1,635,183 | 1,429 | 162016.24 | 7.32 | 11 | 22 | Cell membrane |
| CcABCG28 | KYP54148 | LOC109808608 | 3,046,631–3,056,753 | 1,482 | 167729.07 | 8.19 | 11 | 23 | Cell membrane. Chloroplast. |
| CcABCG29 | KYP54262 | LOC109807460 | 4,014,294–4,034,321 | 1,500 | 170370.83 | 6.75 | 11 | 26 | Cell membrane. Chloroplast. |
| CcABCG30 | KYP55666 | LOC109807699 | 19,989,558–20,004,511 | 1,418 | 161820.65 | 8.9 | 11 | 26 | Cell membrane |
| CcABCG31 | KYP56982 | LOC109809098 | 34,818,776–34,826,902 | 741 | 81702.72 | 9.05 | 11 | 12 | Cell membrane |
| CcABCG32 | KYP57733 | LOC109807641 | 43,015,398–43,024,752 | 1,449 | 164070.17 | 7.67 | 11 | 24 | Cell membrane |
| CcABCG33 | KYP57734 | LOC109809552 | 43,027,240–43,038,856 | 1,445 | 163218.52 | 8.03 | 11 | 24 | Cell membrane |
| CcABCG34 | KYP57934 | LOC109807560 | 45,075,476–45,077,364 | 609 | 68159.95 | 8.65 | 11 | 1 | Cell membrane |
| CcABCG35 | KYP53034 | LOC109810595 | 391,122–395,936 | 681 | 75466.37 | 8.91 | Un | 5 | Cell membrane |
| CcABCG36 | KYP53068 | LOC109810569 | 840,758–848,796 | 1,115 | 123587.99 | 9.16 | Un | 14 | Cell membrane |
| CcABCG37 | KYP52287 | LOC109811196 | 2,121–13,637 | 1,229 | 138171.93 | 8.81 | Un | 20 | Cell membrane |
| CcABCG38 | KYP52347 | LOC109811227 | 785,413–787,392 | 651 | 73181.09 | 8.45 | Un | 1 | Cell membrane |

**Table 1** (*continued*)

| Gene | Accession | Location | Position | Length (aa) | Mw | pI | Chr | Exon | Sub-loc |
|---|---|---|---|---|---|---|---|---|---|
| *CcABCG39* | KYP51780 | LOC109811628 | 147,015–153,318 | 723 | 80384.61 | 8.89 | Un | 12 | Cell membrane |
| *CcABCG40* | KYP51380 | LOC109811918 | 541,176–543,014 | 612 | 68365.89 | 9.39 | Un | 1 | Cell membrane |
| *CcABCG41* | KYP49769 | LOC109813109 | 127,353–133,820 | 683 | 77708.58 | 9.05 | Un | 10 | Cell membrane |
| *CcABCG42* | KYP49445 | LOC109813363 | 17,796–28,760 | 744 | 82372.18 | 8.91 | Un | 12 | Cell membrane |
| *CcABCG43* | KYP48955 | LOC109813737 | 400,317–424,748 | 1,358 | 154173.34 | 8.61 | Un | 21 | Cell membrane |
| *CcABCG44* | KYP48157 | LOC109814303 | 92,212–96,948 | 678 | 75455.58 | 9.06 | Un | 5 | Cell membrane |
| *CcABCG45* | KYP47060 | LOC109815112 | 59,170–61,921 | 755 | 83546.16 | 9.22 | Un | 1 | Cell membrane |
| *CcABCG46* | KYP41536 | LOC109818995 | 187,197–197,961 | 1,445 | 163678.86 | 8.75 | Un | 24 | Cell membrane |
| *CcABCG47* | KYP39219 | LOC109788852 | 72,316–76,120 | 624 | 69345.94 | 8.87 | Un | 4 | Cell membrane |
| *CcABCG48* | KYP37548 | LOC109790064 | 6,674–12,941 | 1,087 | 120433.13 | 8.96 | Un | 14 | Cell membrane |
| *CcABCG49* | KYP36564 | LOC109790672 | 14,766–22,348 | 1,457 | 164896.42 | 8.25 | Un | 24 | Cell membrane |
| *CcABCG50* | KYP35274 | LOC109791517 | 44,302–58,642 | 1,473 | 167555.55 | 7.74 | Un | 20 | Cell membrane |
| *CcABCG51* | KYP32835 | LOC109793033 | 5,717–9,684 | 628 | 70573.41 | 8.74 | Un | 4 | Cell membrane |

were localized in the cell membrane. Three ABCG transporters (CcABCG26, 28, 29) were found to be localized in the chloroplast and not just on the cell membranes (Table 1). To confirm our prediction, we verified the subcellular localization of the ABCG transporter by transiently expressing *CcABCG7-eGFP* in tobacco leaves. The results showed that the green fluorescent signal of CcABCG7-eGFP-pROK II was mainly detected in the cell membrane (Fig. S2).

## Analysis of motifs and conserved domain of CcABCGs

To identify the structure and function of the ABCGs in the pigeon pea, motif analysis was performed based on the phylogenetic analysis (Fig. 3A). A total of 10 conserved motifs were predicted and the width of those motifs ranged from 32 to 123 amino acids. The number of motifs in each amino acid varied from 2 to 10. Results showed that the number of motifs in the WBC was significantly less than the number of motifs detected in the PDR. All 10 motifs were annotated using the InterPro program to identify the motif structure. Annotation analysis demonstrated that Motifs 3, 7, 9, and 10 did not annotate at all; Motif 6 and Motif 8 annotated in the ABC_2_trans structure (IPR013525); Motif 1,2,4,5 annotated in the P_loop_NTPase structure (IPR027417); Motifs 1, 4, and 5 were annotated in the ABC_transporter_like structure (IPR003439). Our results indicated that the motifs of the same annotation were relatively conservative in structure, such as Motifs 1, 4, and 5 (Table S3).

We examined the conserved domain of CcABCGs to explore the ABCG transporter's domain function using HMMER servers (Fig. 3B). The HMME model, "ABC2_membrane" indicated the TMD domain, and "ABC_tran" indicated the NBD domain. The results showed that the ABCG transporters were different from other members of the ABC transporter family with an inverted "TMD-NBD" domain arrangement pattern. All ABCG transporters contained the NBD domain, but several ABCG transporters did not contain the TMD domain. The WBC subgroup contained one NBD domain with a length that increased from 97 to 153 and one or no TMD domains with a length of

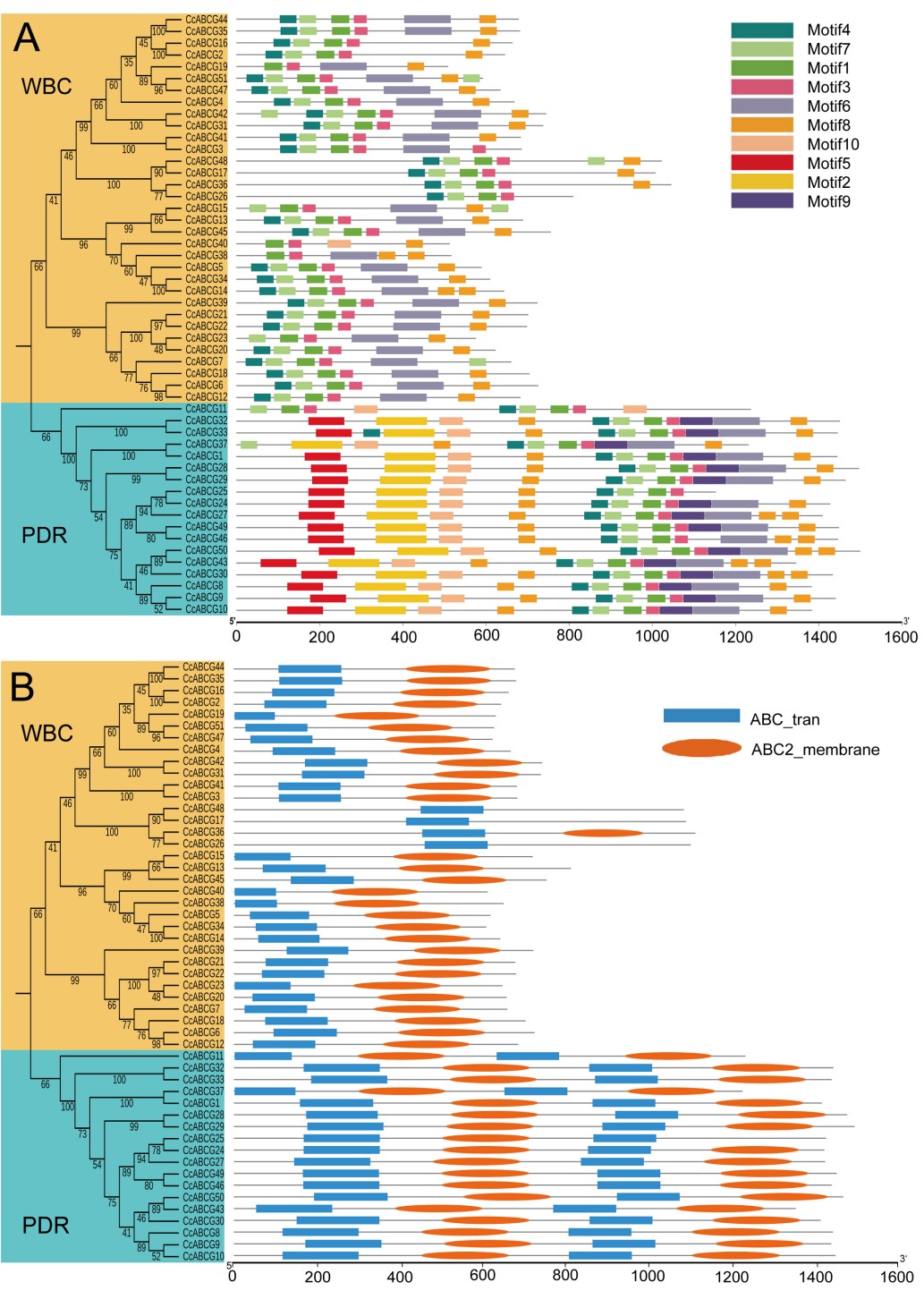

**Figure 3 Motifs and conserved domain of CcABCGs.** (A) The 10 conserved motif of CcABCGs on the MEME software. Motif annotation is shown as color legends. Annotates of motif were listed on Table S3. (B) Conserved domain of CcABCGs. Blue indicates the NBD (nucleotide binding domain), and orange indicates the another conserved domain TMD (trans-membrane domains).

approximately 200, while the PDR subgroup contained two NBD domain with an amino acid length of 138 to 200 and a domain approximately 200 TMD long.

CcABCG35 was selected for homology modeling to explore the molecular functions of CcABCGs. The best template of CcABCG35 was 6hij.1A (Seq Identity:33.05%, GMQE:0.58, Coverage:0.85) using the Swiss-model program. According to the PROCHECK evaluation results, Ramachandran Plot results in the model evaluation revealed that in each ABCG model >90% of the amino acid residues were distributed in the allowed region, indicating that the quality of the CcABCG protein model obtained by homology modeling was reliable (Fig. S3A). A model of CcABCG35 revealed that Walker A and Walker B were located in the NBD domain, while the TMD domain contained six alpha helices (Fig. S3B).

## Gene structure and cis-elements analysis of CcABCGs

Gene structure was analyzed based on the phylogenetic relationship of the ABCG gene family to better understand their structural evolution (Fig. 4A). The highly conserved exon sequence is an essential sequence for the ABCG transporter to perform gene functions, and the differences in introns may be based on dissimilar regulatory mechanisms for the existence of genes. The results indicated that all ABCG transporters in the pigeon pea contained different numbers of exons and introns. The number of exons in *CcPDRs* was significantly higher than that in *CcWBCs*. The number of exons of *CcPDRs* ranged from 5 to 25, while the number of exons of *CcWBCs* varied from 1 to 14 and there were large differences in the number of introns. There were approximately 20 introns in *CcPDRs* and about 5 in *CcWBCs*.

A 2,000 bp region upstream of the promoter was selected for analysis of the cis-acting element of the pigeon pea ABCG gene family in order to explore the expression elements of the promoter region (Fig. 4B). Three cis-acting elements were screened, and focused on hormone-responsive elements, light-responsive elements, and stress-responsive elements. The number of light-responsive elements in the promoter region of the pigeon pea ABCG transporter was found to be relatively large. Hormonal response elements had a large number of copies in the promoter region of the ABCG transporter, including auxin response elements (AuxRR-core, TGA-element, etc.) and gibberellin response elements. There were fewer stress response elements than hormone response elements and some genes did not contain stress response elements (*CcABCG14*, *CcABCG10*, etc.). Drought-related cis-acting elements were identified simultaneously in *CcABCG7* and *CcABCG24*. Low-temperature stress-related cis-acting elements (LTR) were identified in *CcABCG28* and their elements were highly conserved with short sequences composed of CCGAAA.

## Go annotation and function prediction

The *ABCG* gene in pigeon pea was annotated and divided into three categories: molecular function, biological processes, and cell components. The annotation results showed that all 50 genes were annotated into three major categories, but CcABCG48 did not show results from the blast alignment method and UniProtKB ID and all other usable data are provided in Data S5. The transmembrane transport (GO:0055085) had the highest

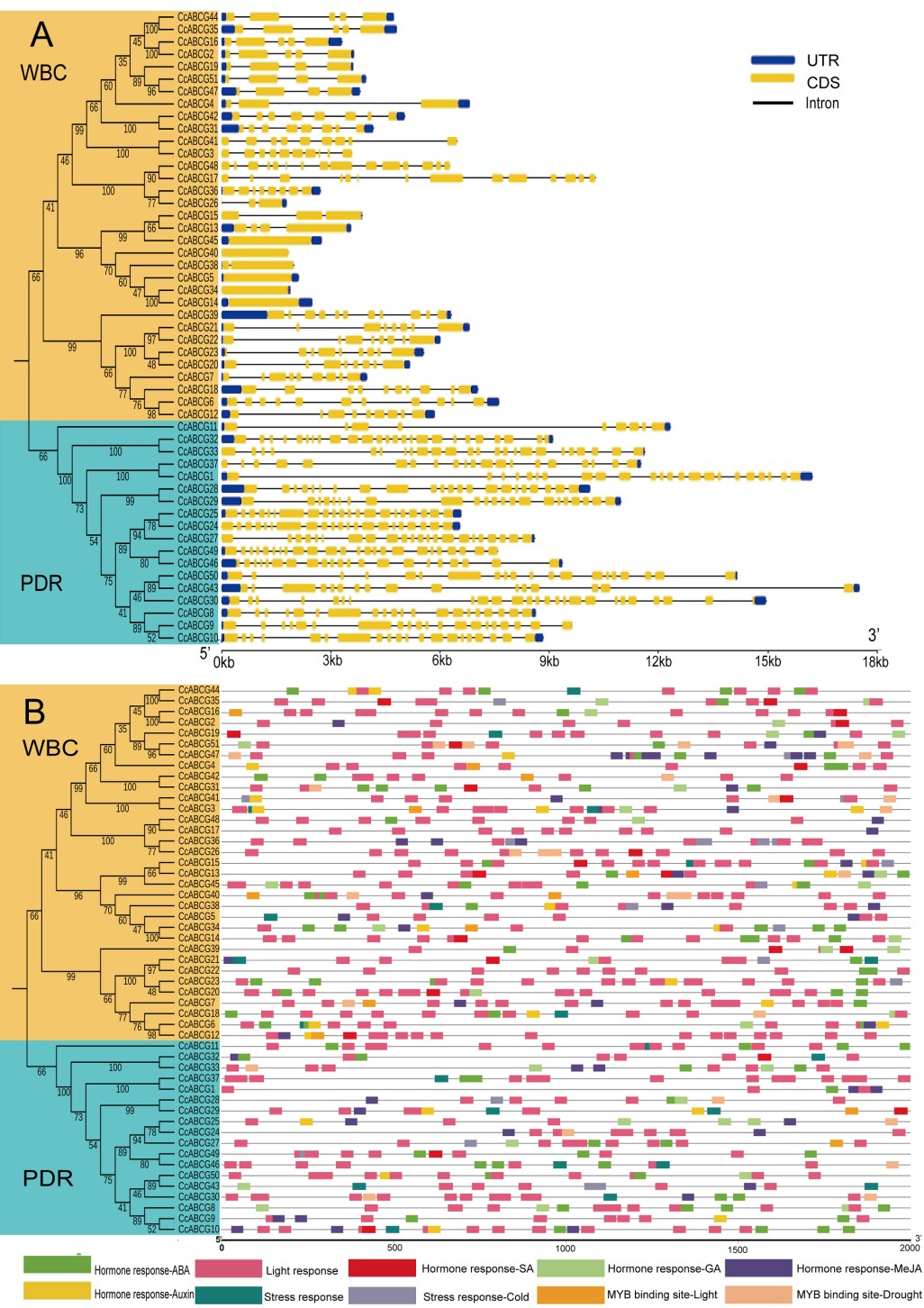

**Figure 4  Gene structure and cis-elements analysis of *ABCG* genes.** (A) Intron-exon structures of *CcABCG* genes. Yellow rectangles: coding sequences (CDSs); thin lines: introns; blue rectangles: untranslated regions (UTRs). (B) Putative regulatory cis-elements in the *ABCG* gene promoters. The relative positions of elements are labeled with capital letters in the figure.

prominence and 45 genes were annotated in the transmembrane transport. CcABCG28 and CcABCG29 were blasted in AB36G_ARATH (UniProtKB ID), and annotated in 42 GO annotations, which included "response to salt stress" (GO:0009651) and "root development" (GO:0048364). Fifty *ABCG* genes were annotated to the broad category of molecular functions related to ATPase activity and ATP binding. The major category of cellular components found most of the *ABCG* genes to be annotated on the membrane (Fig. 5).

## Expression analysis of the pigeon pea ABCG gene family in different organs

To explore the expression pattern of the ABCG gene in different organs during the growth and development of pigeon peas, the organs (roots, stems, leaves, and flowers) of one-year-old pigeon peas were collected to extract RNA for qRT-PCR analysis for the expression of the ABCG genes. Tissue-specific expression analysis found that the expression of ABCG transporters in different tissues was not significantly different. However, there was an obvious difference in the expression level of the ABCG transporter gene in flowers, which was significantly lower than that in the roots, stems, and leaves. The expression of *CcABCG24* was significantly higher than that of any other gene (Fig. 6).

## Expression analysis of the pigeon pea ABCG transporters under different abiotic stresses

To explore the expression level of the pigeon pea *ABCG* gene family under different abiotic stresses, two-week-old pigeon pea seedlings were selected and transplanted into the culture environments at 4 °C, 42 °C, 200 nmol/L NaCl, 100 μmol/L AlCl₃ and 250 nmol/L mannitol in solid MS medium, respectively. *CcABCG5,7,14,19,21* belonged to the WBC subgroup, and *CcABCG10,24,28,29,32* belonged to the PDR subfamily. We found that the expression difference of 10 genes in the roots was significantly higher than in the leaves (Fig. 7).

The two major subfamilies of ABCG transporters had significant differences in expression in the roots (Figs. 7A, 7C, 7E, 7G and 7I) under 4 °C stress and NaCl stress. The *PDR* family genes were significantly up-regulated under both stressors. *CcABCG24* was up-regulated 40-fold after 12 h at 4 °C and *CcABCG28* increased 65 times. Under the initial NaCl treatment, *CcABCG24* increased 10 times. However, it is worth noting that with an increase of the treatment time, the expression of *CcABCG24* decreased to below the initial level at 12 h. We found that the relative expression changes of the WBC subgroups under drought and aluminum stress treatments were opposite, as shown in Figs. 7E and 7G. The expression level of *CcABCG5* increased under drought stress at 6 h but did not change significantly under aluminum stress. *CcABCG19* also exhibited the same pattern of opposite expression in the treatments of mannitol stress and AlCl₃ stress. Under high-temperature stress, we observed that the expression of *CcABCG7* was down-regulated but the expression of *CcABCG7* in leaves was up-regulated.

There was a relatively mild difference in the expression levels in leaves (Figs. 7B, 7D, 7F, 7H and 7J). *CcABCG28* experienced up-regulation under 4 °C treatment. However, *CcABCG7* was significantly up-regulated after 6 h of drought stress. In the same situation,

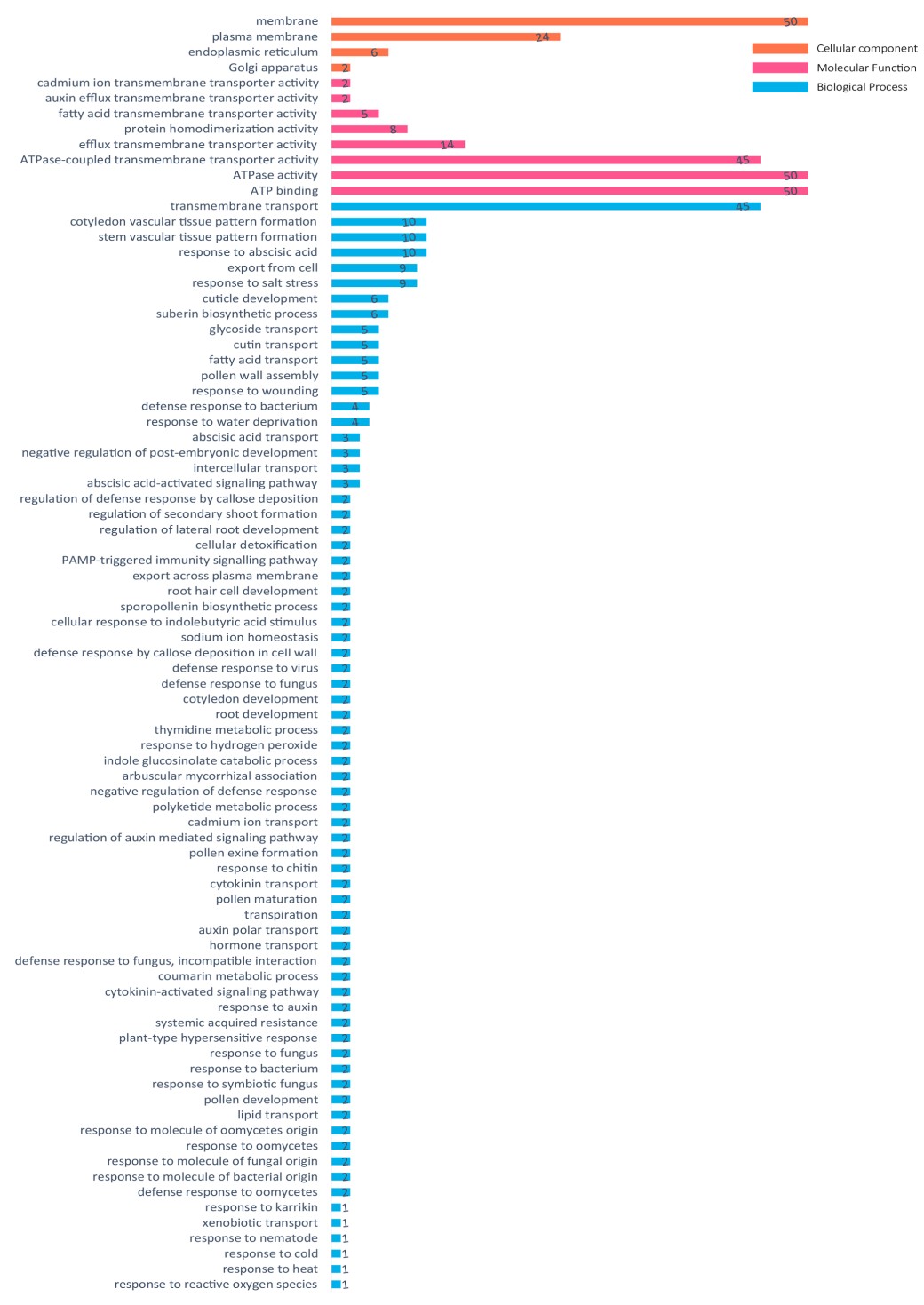

**Figure 5** **Go annotation of ABCG transporters in pigeon pea.** GO annotations of ABCG transporters in pigeon pea were predicted using Swiss-prot database serve. Red means molecular function, yellow means cell component and blue means biological process. Displaying only results for FDR *P* < 0.05.

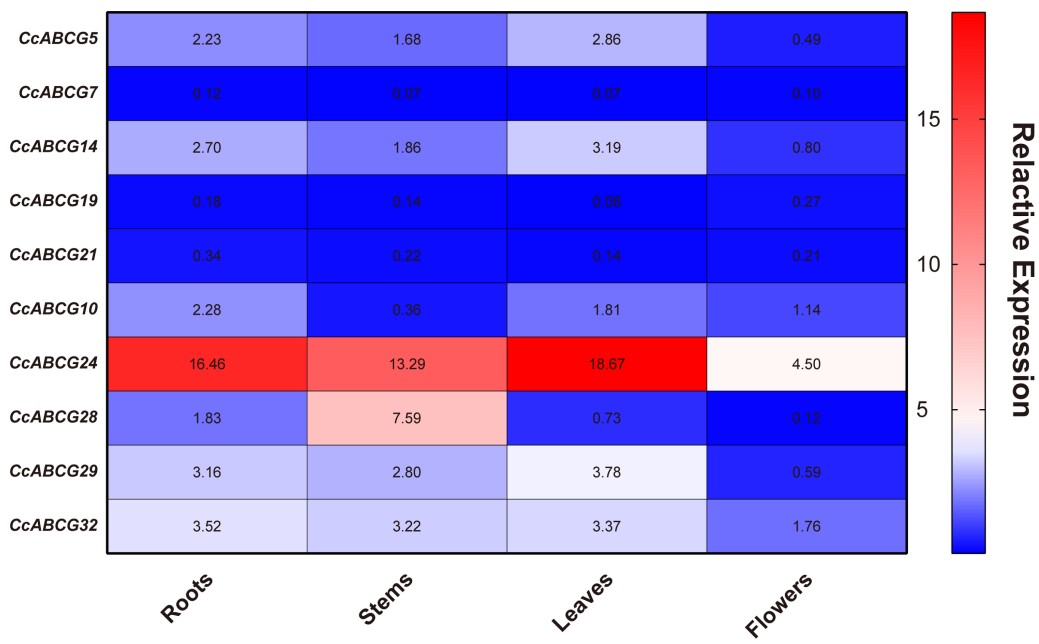

**Figure 6** **Expression analysis of pigeon pea *ABCG* gene in different organs.** The expression levels of 10 genes in different tissues are shown in the heat map. Three biological replicates per sample. Values represent means ± SEM.

*CcABCG7* was up-regulated 11.3 times after 12 h of aluminum stress treatment. However, under sodium stress, *CcABCG7, 19, 21* were down-regulated. The ABCG transporter was found to have different expressions in response to different environmental stresses.

## DISCUSSION

ABC transporters are found in animals and plants and have a large number of complex biological functions. A total of 155 ABC transporters were identified in the pigeon pea genome, which were divided into 8 subgroups of ABCA-ABCI (Fig. S1). Among them, the ABCE and ABCF subfamily had no transport function because their proteins were localized to the endoplasmic reticulum without a transporter region. The ABCB subfamily may be involved in the transport of auxins, secondary metabolites, and heavy metal salts (*Kang et al., 2011*; *Verrier et al., 2008*). The ABCC subfamily may be involved in plant chlorophyll transport and cell detoxification among other functions (*Hashimoto & Yamada, 2003*). The ABCG transporters contained 51 members and was the largest group of ABC transporters in the pigeon pea, which was larger than reported for *Arabidopsis* (44) and rice (50), but smaller than the 116 members of ABCG reported in soybean (*Mishra et al., 2019*; *Jasinski et al., 2003*).

As shown in Fig. 1, the ABCG subfamily could be further divided into a full-molecular transporter PDRs and a semi-molecular transporter WBCs, in which PDRs contained 18 ABCG transporters and WBCs included 33 ABCG transporters. The conserved domain of the ABCG transporter is composed of NBD and TMD, and the ABCG transporter is

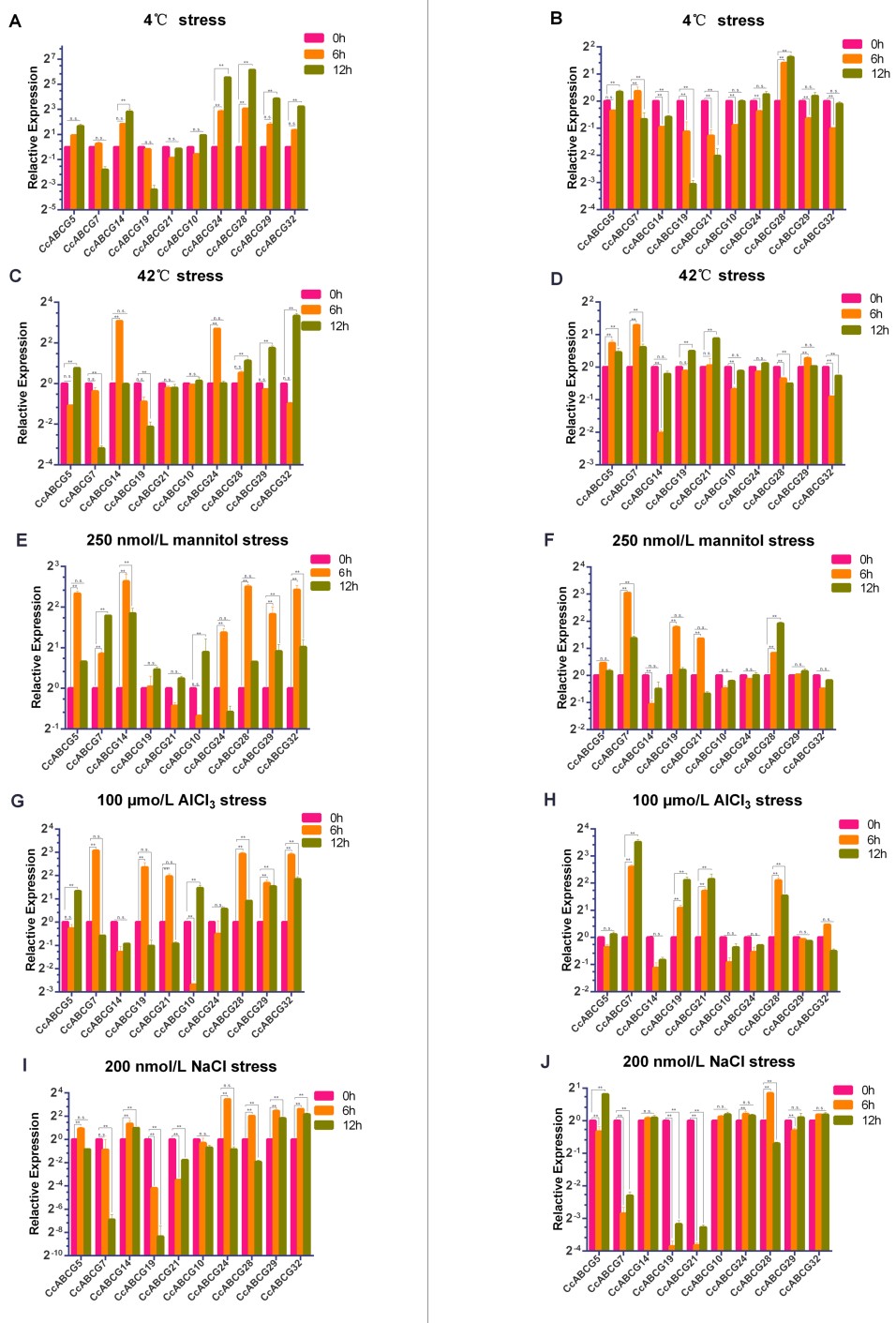

**Figure 7** **Relative expression analysis of the pigeon pea *ABCG* genes under different abiotic stresses.**
Relative expression level of the *ABCG* genes in pigeon pea under different abiotic stresses of 4 °C (A, B), 42 °C (C, D), 250 nmol/L mannitol (E, F), 100 µmo/L AlCl₃ (G, H), 200 nmol/L NaCl (I, J). Expression analysis was performed using a relative quantitative method $2^{-\Delta\Delta Cq}$, and *CcActin* as internal reference gene. Relative expression level of roots: A, C, E, G, I; relative expression level of leaves: B, D, F, H, J. Three biological replicates of each sample. Values represent means ± SEM. Asterisks indicate significant difference as determined by Dunnett's multiple comparisons test (**, $P < 0.01$). n.s., no significant difference.

different from the conserved domain arrangement of other ABC transporter subfamilies. The conserved domain composition of the ABCG transporter is a trans-TMD-NBD structure (Fig. 3) (*Van den Brule & Smart, 2002*). The arrangement of ABC domains and their transmembrane domains are highly conserved, while the number of transmembrane helixes and their arrangements is not necessarily conserved, and determines their functional differences (*Andolfo et al., 2015*; *Goodman, Casati & Walbot, 2004*; *Locher, 2004*). ABCG transporters are involved in many physiological activities of plants, including the transport of small molecular compounds, secondary metabolites, and are active in disease resistance, hormonal regulation, and adaptation to changes in the external environment (*Alejandro et al., 2012*; *Bird et al., 2007*; *Lee et al., 2010*). Arabidopsis cutin and wax secretion require the participation of *AtWBC11* (*Panikashvili et al., 2007a*). Rice *OsPDR9* could be induced by methyl jasmonate, and *AtPDR8* could be induced by salicylic acid (*Kim et al., 2007*; *Kuromori et al., 2010*; *Moons, 2008*). External stressors on plants cause the signal receptors on plant cells to first sense the external stress signal and produce a second messenger transmitted inside the cell, such as $Ca^{2+}$, reactive oxygen and ABA. The stress response also mediates proteins, causing phosphorylation to activate downstream transcription factors and stress-related target genes, thereby resisting the destruction of plant cells (*Mittler, Finka & Goloubinoff, 2012*; *Farooq et al., 2009*). Many *ABCG* genes annotated hormone transport and regulation-related functions (Fig. 4B and 5). *AtABCG22* was induced by drought stress, possibly by affecting the stomata and increasing transpiration (*Kuromori, Sugimoto & Shinozaki, 2011*). In our study, *CcABCG28* (PDR) may be involved in the response to abiotic stresses including colder temperatures and salinity, while *CcABCG7* (WBC) in the pigeon pea tended to respond to drought and aluminum stress, as shown in Fig. 7.

The analysis of the location of the ABCG family in the pigeon pea chromosome found that most of the *ABCG* gene was located on chromosome 11, as shown in Fig. 2. The major part of most plant genomes consisted of different repeating DNA elements. These sequence elements are essential for the large-scale organization and evolution of the plant genome (*Kubis, Schmidt & Heslop-Harrison, 1998*). Our results showed that there were two pairs of paralogous genes (*CcABCG8/9, CcABCG32/33*) closely linked to chromosome 3 and chromosome 11, and the tandem replication led to the expansion of these two genes. Introns are important components in the genome of eukaryotes. The typical 5′-GT...AG- 3′ of intron is an important marker of gene splicing and an important feature of introns in eukaryotic mRNA sequences (*Rose et al., 2016*; *Mukherjee et al., 2018*). We performed exon/intron analysis of the ABCG transporter to determine the stress regulation of the splicing process in Fig. 4A. Our results indicated that PDR contained more intron structures than WBC. It is thought that PDR had more variable splices and functions and may respond more frequently to stress. Furthermore, introns may also affect gene expression, and more introns may have a stronger regulatory effect (*Rose et al., 2016*; *Mukherjee et al., 2018*). There is no evidence supporting the theory that more exons make them prone to more regulation. PDR/WBC have different regulatory effects, which depends on deeper molecular mechanisms. The cis-acting elements are involved in the binding of transcription factors (TF) and regulate the expression of the gene

(*Toledo et al., 2011*). Our study found that there were many low temperature-related elements in the promoter region of the pigeon pea, which might be induced by chilling in the pigeon pea (Fig. 4B). The promoter region of the pigeon pea ABCG transporter had a large number of hormone-regulated expression elements (abscisic acid responsiveness, MeJA responsiveness), which also demonstrated the important role of ABCG transporters in the regulation of plant hormones (*Kuromori et al., 2010*; *Wu, Lewis & Spalding, 2007*). *Kuromori, Sugimoto & Shinozaki (2011)* found that *AtABCG25* is mainly expressed in vascular tissues and can transport ABA out of cells.

ABC transporters are essential for plant development and play a role in the processes of gametogenesis, seed development, seed germination, organ formation, and secondary growth (*Do, Martinoia & Lee, 2018*). We performed expression analysis of the pigeon pea in different organs and under different abiotic stresses (Figs. 6 and 7). The expression levels of all genes in flowers were not overtly different than in other tissues, while *CcABCG24* was more highly expressed in other tissues. *CcABCG24* was expressed in the stems of pigeon peas in significantly higher amounts than in other tissues, showing that ABCG transporters are found in active regions such as trans-membrane proteins. *AtABCG25* is mainly expressed in vascular tissues, can transport ABA to outside of cells (*Kuromori, Sugimoto & Shinozaki, 2011*), and is closely related to the formation of plant vascular bundles. We showed that the expression of most genes in the roots was significantly higher than that in the leaves by analyzing the expression levels of the pigeon seedlings under different stresses. Interestingly, the expression of *CcABCG28* is up-regulated, regardless of any tissue (root or leaf), especially in the leaves. The up-regulation of this expression is significantly higher than any other gene under low-temperature stress. At present, the regulation mechanism of the *CcABCG28* homologous gene in other species is still unclear, but it was shown to play an important role in the growth and development of the pigeon pea. We also found that *CcABCG24* was highly expressed under NaCl treatment, and its homologous gene *AtPDR12* was of great significance for Pb(II) resistance in Arabidopsis (*Lee et al., 2005*). *CcABCG24* played an important role in response to salt stress in this experiment. In response to drought and salt stress in plants, the cuticle lipid coding ABCG transporter gene was significantly up-regulated, indicating that the ABCG transporter has an important role in adapting plants to drought and saline-alkali environments (*Luo et al., 2007*; *Panikashvili et al., 2007a*). The expression level of *CcABCG7* was up-regulated in pigeon pea roots under drought and aluminum stress, which proved its important role in stress tolerance.

## CONCLUSIONS

A total of 51 ABCG transporters were identified and divided into two subgroups: WBC and PDR. We analyzed the protein structure and gene structure cis-elements on the pigeon pea ABCG transporters. The highly conserved NBD domain determines the important function of the ABCG transporter. *CcABCG28* was significantly up-regulated under low temperature stress, while *CcABCG7* responded to drought stress. In conclusion, our results reveal the role of ABCG transporters in abiotic stress resistance and broaden the research direction in abiotic stress resistance of pigeon peas.

## ACKNOWLEDGEMENTS

The authors gratefully acknowledge the support of The College of Forestry, Beijing Forestry University. and Beijing Advanced Innovation Center for Tree Breeding by Molecular Design.

### Funding

This work was supported support by the Beijing Forestry University New Teachers Scientific Research Startup Fund Project (BLX201807), the National Natural Science Foundation of China (31930076), Outstanding Young Talent Fund in Beijing Forestry University (2019JQ03009), the National Natural Science Foundation of China (31901281), National Key R&D Program of China (2018YFD1000602, 2019YFD1000605-1), the China Postdoctoral Science Foundation (2019M660505). The funders had no role in study design, data collection and analysis, decision to publish, or preparation of the manuscript.

### Grant Disclosures

The following grant information was disclosed by the authors:
Beijing Forestry University New Teachers Scientific Research Startup Fund Project: BLX201807.
National Natural Science Foundation of China: 31930076.
Outstanding Young Talent Fund in Beijing Forestry University: 2019JQ03009.
National Natural Science Foundation of China: 31901281.
National Key R&D Program of China: 2018YFD1000602, 2019YFD1000605-1.
China Postdoctoral Science Foundation: 2019M660505.

### Competing Interests

The authors declare there are no competing interests.

### Author Contributions

- Lili Niu analyzed the data, prepared figures and/or tables, authored or reviewed drafts of the paper, software, and approved the final draft.
- Hanghang Li conceived and designed the experiments, performed the experiments, analyzed the data, prepared figures and/or tables, authored or reviewed drafts of the paper, and approved the final draft.
- Zhihua Song performed the experiments, prepared figures and/or tables, and approved the final draft.
- Biying Dong conceived and designed the experiments, prepared figures and/or tables, and approved the final draft.
- Hongyan Cao conceived and designed the experiments, performed the experiments, prepared figures and/or tables, and approved the final draft.

![PeerJ](PeerJ logo)

- Tengyue Liu performed the experiments, authored or reviewed drafts of the paper, and approved the final draft.
- Tingting Du, Wanlong Yang and Rohul Amin analyzed the data, authored or reviewed drafts of the paper, and approved the final draft.
- Litao Wang analyzed the data, prepared figures and/or tables, and approved the final draft.
- Qing Yang analyzed the data, prepared figures and/or tables, authored or reviewed drafts of the paper, and approved the final draft.
- Dong Meng and Yujie Fu conceived and designed the experiments, authored or reviewed drafts of the paper, and approved the final draft.

## Data Availability

Raw measurements are available in the Supplemental Files.

## Supplemental Information

Supplemental information for this article can be found online at http://dx.doi.org/10.7717/peerj.10688#supplemental-information.

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
