# Peer review of "The functional analysis of ABCG transporters in the adaptation of pigeon pea (Cajanus cajan) to abiotic stresses"

_PeerJ, doi:10.7717/peerj.10688_

## Round 0.1 · original submission · Major Revisions

Please address the concerns of all reviewers and revise your manuscript accordingly.

Reviewer 1 ·

Basic reporting

Abstract:
External environment should be replaced by ‘adverse environment’
Keywords were missing
Line 44, remove ‘while’
Introduction
1. The ATP-binding cassette subfamily G is the largest subfamily of ABC transporters family. It was found that ABCG protein was widely distributed in plants and participated in many key physiological processes. Need reference.
2. Line 59, ‘including’ should be replaced by ‘includes’
3. Line 61, ‘with’ should be added before ‘approximately 120 amino’
4. Line 83, it – It
5. Line 80-83, 85-86 reference
6. Line 91-98 mainly described the result you have obtained, which is not suitable to put in the section of Introduction. Please add your hypothesis of the study.
M & M
1. The seeds were sown and after 1 year, the samples were taken. The greenhouse information you provided is far more than enough since the paper only mention the photoperiod of 16h/25°C day and 8h/16°C night. This is not possible that during whole year, the temperature is so stable and the light length as well. Another question is the light level, humidity and irrigation system.
2. ‘One years’ to be replaced by ‘one year’
3. ‘A incubators’ to be replaced by ‘incubators’
4. why did you choose 6 h and 12 h after stress?
5. Why is it 1 year after growth that you took the sample
Discussion
1. Was the ABCG the largest group in rice and Arabidopsis?
2. Fig 2 was not discussed indicating it is less important?
3. In our study, CcPDRs may be involved in the response of pigeon pea to abiotic stresses such as drought, chilling, salt, while CcWBCs had the ability to deal with high temperature, drought and aluminum stress as show in Fig.5B. I am confused about which belongs to CcPDRs and which was CcWBCs in Fig. 5B.
4. Line 317-321. Long sentence that need to divide. Moreover, it need reference to support. The same for line 323-328.
5. Line 329-346 need more and deeper discussion. For instance, any proof from previous studies indicated that the expression of the ABC in different organs were diverse. The up-regulation of the genes at abiotic stresses corresponded to the previous studies or not and why?
6. The discussion need to deeply discussed by combining with previous publications to improve the depth of the paper.
Conclusion
The results were missed since it only mentioned what you did.
Reference
Make the format corresponding to the requirement from the PeerJ.

Experimental design

The experimental design are ok even though there were something remaining unclear.

Validity of the findings

The results were validated by qRT-PCR. Even though this validation is fundamental, the author did show some proof from the result of qRT-PCR.

Reviewer 2 ·

Basic reporting

no comment

Experimental design

no comment

Validity of the findings

no comment

Additional comments

1. The authors should work on the language. The entire article suffers from this one way or another.

2. line 93:ABCG transporters...distributed the cell membrane. What does it mean?

3. line 221: The authors mentioned that they performed subcellular localization. It should be termed as a prediction of subcellular localization.

4. line 223: The authors should clarify the CcABCGs are predicted to be localized in the cell membrane.

5. line 225: The statement that the ATP produced in photosynthesis provides energy for ABCG transport is not supported by any fact. Just because something is predicted to be localized in the chloroplast doesn't mean a direct correlation between the ATP usage and transport through the channel.

6. line 238-242: Additional information regarding the homology modeling is required. What was the starting model? How much sequence/structural similarity between the source and the query? Also, what do we learn from this? The authors mentioned only the number of TMD and NBD domains in the proteins as an outcome of the modeling experiments. Further analysis of the models could shed light on the structure-function relationship of these proteins and may make the manuscript significantly stronger.

7. line 243-250: The authors reported the analysis of the exon/intron composition of the genes. What is the significance of this with respect to the paper? How this is related to the abiotic stress response of these gene-products?

8. line 257-259: The observation of the presence of the low-temperature related elements should be investigated in detail. Are the sequences of these elements very conserved? How about between different plants?

9. line 273-274: The word almost significant could be replaced with proper statistical numbers and significance.

10. line 317: The authors stated that the promoter analysis were done in Fig. 3. There are no promoter analysis data reported in Fig. 3.

11. Each and every figure legends should be rewritten. Figure legends should be self-sufficient and must contain all information to decipher the figure.

12. Figure 2: The location of the genes in the respective chromosomes were shown meticulously. However, how is this information central to the paper? This should be detailed in the text.

13. Figure 3A: Are the sequences of the specific domains really necessary to be inside the figure? Are these sequences absolute or just consensus? If they are consensus, how were they obtained? How much differences are there between the domains in different proteins?

14. Figure 3B: Macromolecule structures visually look better with a white background. I am still not clear what are these two models telling us about this story? Figure legends should be more descriptive.

15. Figure 4A: What this have to do in the paper? Does this figure contribute in any way towards validating the findings/main story? Please clarify.

16. Figure 5B: What is CK? Error bars have been used -- are they SD or SEM? How were they calculated? Needs to be mentioned in the text.

17. Figure 5B: Are the differences that are seen here significant? Simple statistical analysis should be done on the numbers and always reported with this type of experiment.

18. This article needs a proper introduction to various types of abiotic stresses and their importance. Why used AlCl3? Why is that a key chemical deserved to be tested? Pertinent information like this is a prerequisite for a complete story.


The authors have done a significant amount of work for this article. However, the results were ineptly written and presented. In my view, the authors tried to report many information that are not related directly to the story and thereby diluting the core message. Also, the parts which could have made the manuscript stronger were unfortunately either poorly analyzed or negligently reported.

Reviewer 3 ·

Basic reporting

no comment

Experimental design

no comment

Validity of the findings

no comment

Additional comments

The authors present a bioinformatics and expression analysis of the ABCG transporters in pigeon pea (Cajanus cajan).
1. The writing and organization of the paper MUST be improved.
line 42: ABC transporters plays…?
line 47: was that the AtABCB1 (also known as AtMDR1) was cloned in Arabidopsis in 1992? line 56: According to their structure…?
Line 56: ABCG subfamily were…?
Line 58: while PDRs belongs to…?
Line 60: The NBDs domain consists of…?
Line 64: the hydrolyzes reaction of …?
Line 68: more than 40 ABCG transporters members…?
Line 73: “the plant's tolerance salt and drought stress…” should be “the plant's tolerance (to) salt and drought stress…”
Line 76-79, line 87-90…
Line 92-93 ABCG transporters in pigeon pea was located on…?
Line 94: classed into..?
Line 96: stress of temperature, drought and high concentration of aluminum…?
…. …. …. ….
I can scarcely read it. Simple grammatical errors can be seen everywhere.

2. In motifs analysis, the authors were identified 10 conserved motifs in the CcABCGs, and these conserved motifs were annotated by InterPro. However, no conserved motifs were annotated as NBD or TMD domains, which were the typical features of ABCG subfamily members.

3. According to the title “ABCG transporters in pigeon pea”, the authors should focus on ABCG transporters in the manuscript. First, the authors should present the recent advances for ABCG transporters in Introduction. Second, the authors should identify the ABC transporter genes in pigeon pea genome, and named as CcABCG1-CcABCG55. And then, genome-wide bioinformatics analysis of the CcABCGs, containing genome distribution, gene duplication, gene structure, evolutionary relationship, conserved motif, promoter analysis and putative expression patterns in different organs and under different abiotic stresses, can be further investigated to assess the importance of these genes in pigeon pea.

4. In method section, the authors claimed that they searched potential ABC proteins using the amino acid sequences of Arabidopsis ABCs as queries. In my opinion, it may omit some ABC proteins. The HMMER program should be better. PF00005: ABC transporter domain; PF01061: ABC-2 transporter domain; PF00664, ABC transporter transmembrane region domain.

5. The subsection “Expression analysis of the pigeon pea ABCG gene family in different organs and under different abiotic stresses” should be divided into two subsections.

6. Why did the authors only choose 10 ABCG genes for qRT-PCR analysis in different tissues and responses to various stresses? Please add more gene information about ABCG genes for qRT-PCR analysis. Segmental and tandem duplication genes may be better.

7. Some references listed in the manuscript are inappropriate. Many recent relevant reports about the ABC transporter family were even not mentioned.

8. The fact that statistical evaluation of the obtained results is missing.

9. The Discussion section does not present any real conclusions and comments. The authors should discuss more about the ABCG transporters in adaptation of pigeon pea to abiotic stresses, and the relationships of CcABCGs and other important ABCG transporters identified in other plants.

---

## Round 0.2 · Major Revisions

Please address remaining critiques of the reviewers and amend your manuscript accordingly.

Reviewer 2 ·

Basic reporting

.

Experimental design

.

Validity of the findings

.

Additional comments

The authors have added new explanations/experimental data in support of their claims and as a rebuttal to the reviewer’s comments. I am satisfied with most of the replies and additions the authors made. However, I still think the authors could concentrate the manuscript more on the effect of stress and gene regulation, rather than the chromosomal positions or intron-exon composition of the genes (authors have dedicated entire figures on them). The argument that having more exon make them prone to more regulation is theoretical and no evidence was provided in support of that statement. It is true that we are still learning new things about the mechanism and regulation of the splicing process and the statement could be theoretically sound but that still doesn’t carry any weight towards the conclusions of the paper.

This manuscript needs to be corrected thoroughly for linguistic issues (grammatical mistakes, sentence structures, typos, etc.) and rewritten for scientific clarity.

Reviewer 3 ·

Basic reporting

No comment

Experimental design

No comment

Validity of the findings

No comment

Additional comments

Thanks for the author's serious response. This revised manuscript has been substantially improved, and the questions that the authors have explained and the errors that the authors have corrected. However, there are still a number of grammatical mistakes, such as line 50, 240, 297, 341, 350, 391, 407, and so on. The authors should seek an English native speaker to correct the grammar mistakes for improving the readability. In addition, the authors should update the descriptions in panels (A-D, G and H) of figure 6. These errors should be addressed before publication.

---

## Round 0.3 · Minor Revisions

It was pointed out by the Section Editor, that the focus of this study is to add to the knowledgebase of the ABCG subfamily in regard to abiotic stresses, yet this appears very limited in scope. There are 51 sequences described and some information provided regarding their function in relation to the basic four stresses tested, yet the meaningful annotations are not presented in a form that can carry this work forward.

Annotations in regards to localization and function should be annotated in ontology terms so that this information can be used in comparison to other systems. Perhaps GO: numerical designation can be added to the supplement table in regard to tissue, functional, and molecular classifications.

Journal manuscripts are often scanned by text-mining software that locates and extracts core data elements, like gene function. Adding standard ontology terms, such as the Gene Ontology (GO, geneontology.org) or others from the OBO foundry (obofoundry.org) can enhance the recognition of your contribution and description. This will also make human curation of literature easier and more accurate. None of this was visible.

The authors speak of laying foundations for future work, yet the readers still are required to re-do the analyses and are not given a baseline to continue this effort, as the current authors do not propose any strategies in moving the current work forward, and much still appears to reply on prediction through software tools than brute force laboratory testing, except for the few attempted.

Based on these considerations, a moderate revision is needed at this stage.

---

## Round 0.4 · Minor Revisions

It is indicated by the reviewer that there are many linguistic issues (sentence formation, typos, grammatical errors, etc.) throughout this draft. Therefore I am strongly recommending that you use some professional writing service before resubmitting this manuscript again.

Reviewer 2 ·

Basic reporting

.

Experimental design

.

Validity of the findings

.

Additional comments

I am scientifically satisfied with the new changes. In terms of language, this manuscript is unfortunately still far away from perfection. There are typos and grammatical errors everywhere. Here are some (corrected) examples just to name a few--

Missing s for third-person singular subjects:
The NBD domain consists of three highly.., NBD domain on the cell membrane hydrolyzes ATP.., etc.

Oxford commas are missing:
In order to reveal the important role of ABCG transporters in resisting environmental stress in pigeon peas, identification, phylogenetic analysis, and expression analysis, etc.

Was/were issues:
gene localization was annotated..., Gene structure was analyzed based..., etc.

Missing auxiliary verbs:
Results have shown that the number of motifs in the WBC...

Is/are issues: The cis-acting elements are...

Typos: gametogenesis, ABCE and ABCF subfamily had no transport function..., etc.


Overall, The manuscript is ready to be published but only after thorough proofreading. I would recommend the authors to use some professional writing service before the final publication.

---

## Round 0.5 · Minor Revisions

While looking at your manuscript, the Section Editor, Gerard Lazo, said: "The authors have partially addressed some of the concerns, but have yet failed to create clear connections. There are no GO: annotations listed with the description terms and one of the description terms was altered. I have taken the liberty to generate the needed annotation, but cannot connect the annotations to any sequences based on the manuscript text; this is needed. It is preferable that usable data be generated in text form rather than in solely in figures. While many figures may look impressive, they are basically useless unless there is hard data provided to back them up. This manuscript requires a bit more clarity in connecting data to the observations, thus requiring additional revision. With eight authors I would think there would be a better division of labor in generating a worthy manuscript.

Listed are GO: annotations painfully extracted from the figure of the manuscript:
+-------+-----------------+--------------------+------------+
| go | name | type | accession |
+-------+-----------------+--------------------+------------+
| 5765 | nitrogen compound metabolic process | biological_process | GO:0006807 |
| 35127 | abscisic acid transport | biological_process | GO:0080168 |
| 35014 | cutin transport | biological_process | GO:0080051 |
| 33241 | cellular response to indolebutyric acid stimulus | biological_process | GO:0071366 |
| 8644 | pollen wall assembly | biological_process | GO:0010208 |
| 8774 | suberin biosynthetic process | biological_process | GO:0010345 |
| 9009 | cotyledon vascular tissue pattern formation | biological_process | GO:0010588 |
| 8658 | stem vascular tissue pattern formation | biological_process | GO:0010222 |
| 30044 | transmembrane transport | biological_process | GO:0055085 |
| 33241 | cellular response to indolebutyric acid stimulus | biological_process | GO:0071366 |
| 298 | intracellular | cellular_component | GO:0005622 |
| 343 | integral component of membrane | cellular_component | GO:0016021 |
| 42946 | ATPase complex | cellular_component | GO:1904949 |
| 36838 | ATPase dependent transmembrane transport complex | cellular_component | GO:0098533 |
| 41525 | ATPase inhibitor complex | cellular_component | GO:1903503 |
| 4563 | ATP binding | molecular_function | GO:0005524 |
| 5204 | ATPase activity | molecular_function | GO:0016887 |
| 1185 | ATPase activator activity | molecular_function | GO:0001671 |
| 5204 | ATPase activity | molecular_function | GO:0016887 |
| 2978 | ATPase activity, coupled | molecular_function | GO:0042623 |
| 22222 | ATPase activity, coupled to movement of substances | molecular_function | GO:0043492 |
| 7058 | ATPase activity, coupled to transmembrane movement of ions, phosphorylative mechanism | molecular_function | GO:0015662 |
| 23462 | ATPase activity, coupled to transmembrane movement of ions, rotational mechanism | molecular_function | GO:0044769 |
| 3231 | ATPase activity, coupled to transmembrane movement of substances | molecular_function | GO:0042626 |
| 21446 | ATPase activity, uncoupled | molecular_function | GO:0042624 |
| 28248 | ATPase binding | molecular_function | GO:0051117 |
| 21447 | ATPase coupled ion transmembrane transporter activity | molecular_function | GO:0042625 |
| 37494 | ATPase coupled ion transmembrane transporter activity involved in regulation of postsynaptic membrane potential | molecular_function | GO:0099581 |
| 37435 | ATPase coupled ion transmembrane transporter activity involved in regulation of presynaptic membrane potential | molecular_function | GO:0099521 |
| 20914 | ATPase inhibitor activity | molecular_function | GO:0042030 |
| 30671 | ATPase regulator activity | molecular_function | GO:0060590 |
| 37660 | ATPase-coupled alkylphosphonate transmembrane transporter activity | molecular_function | GO:0102017 |
| 21991 | ATPase-coupled anion transmembrane transporter activity | molecular_function | GO:0043225 |
| 10076 | ATPase-coupled arsenite transmembrane transporter activity | molecular_function | GO:0015446 |
| 39304 | ATPase-coupled doxorubicin transmembrane transporter activity | molecular_function | GO:1901242 |
| 39572 | ATPase-coupled lipo-chitin oligosaccharide transmembrane transporter activity | molecular_function | GO:1901514 |
| 39305 | ATPase-coupled methionine transmembrane transporter activity | molecular_function | GO:1901243 |
| 10041 | ATPase-coupled molybdate transmembrane transporter activity | molecular_function | GO:0015412 |
| 10043 | ATPase-coupled nitrate transmembrane transporter activity | molecular_function | GO:0015414 |
| 10045 | ATPase-coupled organic phosphonate transmembrane transporter activity | molecular_function | GO:0015416 |
| 10044 | ATPase-coupled phosphate ion transmembrane transporter activity | molecular_function | GO:0015415 |
| 10087 | ATPase-coupled protein transmembrane transporter activity | molecular_function | GO:0015462 |
| 10048 | ATPase-coupled sulfate transmembrane transporter activity | molecular_function | GO:0015419 |
| 16130 | ATPase-coupled thiosulfate transmembrane transporter activity | molecular_function | GO:0032146 |
| 39301 | ATPase-coupled tungstate transmembrane transporter activity | molecular_function | GO:1901238 |
| 21447 | ATPase coupled ion transmembrane transporter activity | molecular_function | GO:0042625 |"

---

## Round 0.6 · Minor Revisions

Reviewer 2 indicated that although the manuscript was significantly improved, it still contains numerous linguistic issues and requires careful proofreading.

Reviewer 2 ·

Basic reporting

.

Experimental design

.

Validity of the findings

.

Additional comments

This manuscript has improved a lot. I, however, standby my previous recommendation that the authors may want to use writing services. There are many sentences that require restructuring. For example the first sentence of the Discussion section --

ABC transporters are distributed in animals and plants, but ABC transporters in plants are
characterized by a large number and complex function, such as 129,127 ABC transporters were
identified in Arabidopsis and rice.

I am not certain what message does this carry? These types of sentences are everywhere. I once again would request the authors to proofread the manuscript thoroughly.

Overall, this manuscript is ready to be published but only after thorough proofreading.

---

## Round 0.7 · accepted · Accept

All issues were addressed and I am happy to accept revised manuscript now.